# Working Time Control and Variability in Europe Revisited: Correlations with Health, Sleep, and Well-Being

**DOI:** 10.3390/ijerph192214778

**Published:** 2022-11-10

**Authors:** Nils Backhaus

**Affiliations:** Federal Institute for Occupational Safety and Health (BAuA), Friedrich-Henkel-Weg 1-25, 44149 Dortmund, Germany; backhaus.nils@baua.bund.de

**Keywords:** working time control, working time variability, health, sleep, well-being, Europe, European Working Conditions Survey (EWCS)

## Abstract

Working time control (WTC) and working time variability (WTV) are two important dimensions of working times, especially with regard to the dynamics of irregular working hours in a changing world of work. Both dimensions are closely related, and the terms are sometimes used synonymously. However, a high degree of WTC does not automatically lead to variable or irregular working hours. On the contrary, WTV is often imposed by the employer and does not necessarily occur in conjunction with high WTC. This article gives an overview of different European WTC and WTV regimes using a typological approach. Based on the European Working Conditions Survey 2015 (EWCS, n = 27,607), four employee groups are compared: those with (1) high WTC and high WTV, (2) high WTC and low WTV, (3) low WTC and high WTV, and (4) low WTC and low WTV. Firstly, the analyses aim to assess whether WTC and WTV vary across European countries due to different working time regimes and in different occupational sectors, i.e., hospitality, retail, and health and social work. Secondly, multi-level analyses are used to describe correlations with health (self-rated health, psychosomatic complaints), sleep (sleep problems), and well-being (WHO-5-Scale). The analyses suggest that WTC and WTV differ between European countries: in the northern countries, high WTC/high WTV is most prevalent, whereas low WTV/low WTC is more common in the other countries. As far as employee health and sleep are concerned, high WTV is associated with poor health, i.e., a greater number of psychosomatic health complaints, worse self-rated health status, and more sleep problems. However, the correlation appears to be weaker for psychosomatic health complaints when employees have high WTC. Significant correlations could not be found for WTC. Low WTC and high WTV is more common in occupational sectors in hospitality, retail, and health and social care; however, these occupational sectors show the same correlations regarding health, sleep, and well-being. The analyses indicate that it is crucial to consider WTV and WTC together in order to understand the dynamics of irregular working hours and health.

## 1. Introduction

Over the last few decades, major changes in the world of work have eroded standard working times and led to the emergence of new working time arrangements [1,2]. Economic requirements for leaner, more flexible business processes in a globalized 24/7 economy are being fueled by digitalization [3,4,5,6]. At the same time, employees now have greater expectations and stronger preferences concerning the balance between private and work life, resulting in more working time autonomy and working from home [7,8,9]. Based on these extensive changes, working time control (WTC) and working time variability (WTV) are two crucial dimensions for the irregularity of working and private life [1,7,10,11,12,13,14,15].

Drawing on the most recent (sixth) wave of the European Working Conditions Survey (EWCS) 2015 [16], this article has three main objectives. Firstly, cross-country disparities and similarities with regard to WTC and WTV in Europe are identified. The various welfare state regimes, social and family policies, and industrial relations systems in Europe give rise to differences in working time arrangements between European countries [8,17,18]. Secondly, WTC and WTV are linked to health, sleep problems, and well-being since both dimensions affect employees in various ways. The typology used is therefore based on both dimensions. Thirdly, WTC and WTV are analyzed in different occupational sectors, i.e., hospitality, retail, and health and social work.

### 1.1. Working Time Control (WTC)

WTC is defined as the employee’s degree of autonomy regarding “how, when or where they work” and is influenced by organizational policies and practices [19], p. 188. WTC can be categorized into one of two dimensions, depending on who sets the working time—the employer (low WTC) or the employee (high WTC) [7,10,20]. While high WTC allows employees to meet their personal needs by providing freedom as to their start and end times and when they take days off or holidays, working times are mostly determined by the employer where WTC is low [21].

Low WTC makes it more challenging to adapt working hours to private life [22], which may increase work–life conflicts [7,13,23,24,25]. Externally determined working hours are perceived as stressful and are related to poor physical and psychological health and well-being among employees [7,22,23,24,26,27,28,29,30,31]. By contrast, high WTC allows employees to adjust their working hours to their private lives and leisure activities [14,32,33], increasing the predictability of working times, which can improve the compatibility of family and work [13,15,27]. Moreover, WTC may act as a resource buffering the adverse effects of work demands such as high work intensity on employee health and well-being [13,14,34]. A review by Nijp, Beckers, Geurts, Tucker, and Kompier [12] shows that WTC positively correlates with the quality of work–life balance, health, and well-being. There are significant differences in the extent of WTC between occupational groups [35]. Occupations in the service sector, e.g., hospitality and retail, often have casual or on-demand work arrangements mainly driven by the employer [36]. On the other hand, many knowledge or white-collar workers perform more complex tasks that allow greater autonomy and discretion, including with regard to working hours [37].

However, more and more researchers have also focused on the dark side of high flexibility and WTC over the past few years, see [38] for a review. It is increasingly being debated whether extensive flexibility or autonomy could be “too much of a good thing” [39,40,41,42,43,44]. In contradiction to many theoretical models, WTC and autonomy can—paradoxically—turn from a resource into a demand [41,45]. This often translates into self-endangering behavior such as long working hours, overtime, high workload, irregular working hours, or sickness presenteeism [42,46,47,48,49]. However, it is not a straightforward matter to judge when WTC is a resource or a demand, something that depends considerably on employee characteristics and needs [50].

### 1.2. Working Time Variability (WTV)

WTV reflects the actual fluctuation and dynamics of working time patterns over time. It can also be described as the degree of (in)stability, irregularity, or (in)consistency of working times [51]. WTV is reflected in a number of aspects [52,53,54], e.g.,

whether the duration of daily or weekly working hours is stable (low WTV) or fluctuates (high WTV),whether the number of working days per week is stable (low WTV) or varies (high WTV), andwhether daily start and end times are identical (low WTV) or variable (high WTV).

Systematic reviews show that the findings reached about WTV and health are somewhat inconsistent, e.g., [55]. Some research suggests WTV is associated with poorer mental health [56] or increased mental stress [57,58]. However, the link between WTV and psychosomatic health complaints or burnout is less clear: while some studies report a slight increase in the risk of burnout for employees with high WTV [13,27], others do not find increased burnout risks [57,59]. Further research also suggests a relationship between high WTV and increased sickness absenteeism [60]. Thus, increased WTV in terms of the timing and duration of working hours is associated with disruption to circadian-controlled functions (sleep, digestive system) and potentially psychosomatic health complaints, which shift work research has also identified in cases of biological desynchronization [61]. WTV and irregular schedules are unequally distributed among the workforce. Especially In the hospitality and retail sectors, unstable and irregular work schedules contribute to precarious working conditions [22,30,51,62]. Employees in the helping professions, i.e., health and social care, often face irregular working hours and shift work (high WTV) too [63,64,65].

### 1.3. Working Time Control and Working Time Variability Combined: A Typology

The inconsistent findings about the association between WTV and health suggest other influencing factors need to be considered. Some authors argue that WTV imposed by employers (i.e., low WTC) differs from WTV determined by employees with high WTC [51,53,54,61,66]. Costa et al. [7] describe WTV as “more subjected to company control and decision” (p. 1127), which is particularly the case if the employer imposes the variability. However, WTC and WTV have rarely been investigated together, e.g., [51,53]. This is the starting point for the following research framework.

To analyze WTC and WTV together, a 2 × 2 matrix framework is proposed [51], see Figure 1. The matrix depicts four types of working time arrangements characterized by different levels of WTC and WTV (A.–D.). Employees with high WTC do not necessarily need to work variable schedules. For example, employees with flexitime can still decide to work the same schedule every day or week (high WTC, low WTV; A. in Figure 1). To address this issue, Allen and colleagues [67] differentiated between flexibility availability and flexibility use. In their terminology, flexibility availability equates to high WTC. Employees who make use of their high WTC to work irregular and variable schedules actively use the flexibility it offers (high WTC, high WTV; B. in Figure 1). Low WTC/high WTV is associated with irregular and unstable working schedules determined by the employer (low WTC, high WTV; C. in Figure 1), e.g., shift work or on-call shifts [58]. If the employer determines working times that are fixed and stable, WTC and WTV are both low (D. in Figure 1).

Research shows that WTC can buffer the adverse effects of working time demands [68]. However, some researchers argue high WTV also worsens health if employees with high WTC are responsible for their working hours: Kubo and colleagues [54] report significantly poorer sleep quality and work–life balance for employees with high WTC and high WTV compared to those with high WTC and low WTV. Studies based on large-scale surveys [53,61] also conclude that high WTV has negative effects, even for employees with high WTC, comparable to those of shift workers with high WTV and low WTC.

### 1.4. Research Questions

This article begins with a combined analysis of WTC and WTV based on a large-scale multinational survey (EWCS). In a descriptive analysis, the distribution of WTC and WTV across Europe is described using representative employee data. In view of the various working time regimes across Europe, the comparative analysis of different countries seems essential to understand the mechanisms of and preconditions for WTC and WTV in different regions and welfare state regimes [69,70].

WTC and WTV’s correlations with health, sleep, and well-being are then analyzed. Based on the research that has been done on WTC and employee-oriented flexibility (see Section 1.1), it is concluded that WTC is beneficial for health, sleep, and well-being. With regard to the relationship between WTV and health (see Section 1.2), it is postulated that WTV has no or slightly negative associations with health, sleep, and well-being. The decisive point of the analysis is the interaction of WTC and WTV (see Section 1.3). Research shows WTC can buffer the adverse effects of working time demands [68]. Thus, it is expected that the negative relationship between high WTV and health, sleep, and well-being be stronger when WTC is low. On the other hand, WTV has slightly fewer negative effects if WTC is high, i.e., if WTV is driven by the employee.

Lastly, the correlations between WTC, WTV, and health and well-being are compared between selected occupational sectors that tend to be associated with irregular schedules (see Section 1.2). In this article, employees from the hospitality, retail, and health and social care sectors are contrasted.

## 2. Materials and Methods

This article draws on the sixth wave of the European Working Conditions Survey (EWCS) carried out in 2015 [16]. The dataset covers employees from 35 countries, i.e., the 28 European Union (EU28) member states, including the United Kingdom (UK) before its withdrawal from the European Union in 2020, and seven other European countries outside the European Union (Norway, Switzerland, Albania, Former Yugoslav Republic of Macedonia, Montenegro, Serbia, and Turkey). The basic population was made up of inhabitants of the above-mentioned countries aged 15 or 16 and older who were employed at the time of the survey. Individuals were considered employed if they worked at least one hour for pay or profit in the week preceding the interview. The survey was conducted by means of computer-assisted personal interviews (CAPI) at the respondent’s home with an average duration of 45 min. It comprises a large number of questions, covering physical and psychosocial working conditions, working time, place of work, health, and well-being. For the analyses, the focus is on employees living in the EU28 (including the UK) up to age 65. For the purpose of analyzing WTC, self-employed workers were excluded. Respondents with missing data on one of the dimensions of the model were also removed prior to analysis. Ultimately, the analysis sample consisted of 27,607 respondents.

### 2.1. Independent Variables

WTC was operationalized by the question “How are your working time arrangements set?” (Q42), (Item numbers from EWCS 2015 in brackets) which was answered with four Likert-scaled categories (1 = “They are set by the company/organization”, 2 = “You can choose between several fixed working schedules determined by the company/organization”, 3 = “You can adapt your working hours within certain limits (e.g., flexitime)”, 4 = “Your working hours are entirely determined by yourself”). For our analyses, the variable was coded either as low WTC time arrangements (1–2) or as high WTC (3–4).

WTV was measured with four items (Q39a-d). “Do you work the same number of hours every day?”, “Do you work the same number of days every week?”, “Do you work the same number of hours every week?”, and “Do you work fixed starting and finishing times?” The respondents were able to answer “yes” or “no”. According to Kubo et al. [54], respondents with at least one “no” answer were assigned to the high WTV group; otherwise, they were coded as fixed schedules (low WTV).

### 2.2. Dependent Variables

Health. For the health outcomes, self-rated health was measured using a single item (Q75) “How is your health in general? Would you say it is 1 = ‘very good’, 2 = ‘good’, 3 = ‘fair’, 4 = ‘bad’, 5 = ‘very bad’”. Psychosomatic health complaints were assessed as follows: “Over the last 12 months, did you have any of the following health problems?” Nine psychosomatic health complaints were included in the questionnaire (Q78a-i: hearing problems; skin problems; backache; muscular pains in shoulders, neck, and/or upper limbs (arms, elbows, wrists, hands, etc.); muscular pains in lower limbs (hips, legs, knees, feet, etc.); headaches, eyestrain; injury (ies); anxiety; overall fatigue). Following the guidance given by Franke [71] and Müller, et al. [72], the final score was generated by counting the “yes” answers. The index ranged from 0 (no complaints) to 9 (all complaints).

Sleep. Sleep problems were measured using three items (Q79): “Over the last 12 months, how often did you have any of the following sleep related problems?” (Q79a-c: difficulty falling asleep, waking up repeatedly during the sleep, waking up with a feeling of exhaustion and fatigue). The items could be answered using a 5-point scale (1 = “daily”, 2 = “several times a week”, 3 = “several times a month”, 4 = “less often”, 5 = “never”).

Well-being was assessed on the WHO-5-Scale [18] comprising five statements (Q86a-e). Participants had to indicate for each statement what was the closest to how they had been feeling over the previous two weeks, e.g., “I have felt cheerful and in good spirits” (response categories: 6-point Likert scale from 1 = “all of the time” to 6 = “at no time”).

### 2.3. Control Variables

Several individual variables were taken into account, such as gender (1 = “male”, 2 = “female”), age (1 = “15–29 years”, 2 = “30–44 years”, 3 = “45–54 years”, 4 = “55–65 years”), education level (1 = “low”, 2 = “medium”, 3 = “high”, as defined by the ISCED), marital status (1 = “no spouse in household”, 2 = “spouse in household”), age of youngest child ≤ 18 years in household (1 = “no child in household”, 2 = “13–17 years”, 3 = “7–12 years”, 4 = “≤6 years”), household size (1, 2, 3, 4, 5, 6 persons and more), occupation (International Standardized Classification of Occupations 2008 (ISCO-08), 2-digits), and economic sector (Statistical Classification of Economic Activities in the European Community (NACE), 2nd revision, 2 digits). Additionally, specific aspects of working time were controlled. Firstly, the actual working time duration was included (Q24: “How many hours do you usually work per week in your main paid job?”). The number of hours per week was coded into six groups (1 = “1–19 h”, 2 = “20–34 h”, 3 = “35–39 h”, 4 = “40–47 h”, 5 = “48–59 h”, 6 = “≥60 h”). Secondly, shift work was considered (Q39e: “Do you work shifts?” Response categories: 1 = “no”, 2 = “yes”). Table 1 gives the sample statistics for all variables.

### 2.4. Data Analysis

Descriptive analyses were conducted with IBM SPSS Statistics 28. Data were weighted using national weights (for country comparison) or overall EU28 weights (for the overall statistics). In order to take the nested structure of the data into account (individuals clustered in countries), linear mixed-effects models were fitted using the R package lme4 [73]. Countries were included as random intercepts (1|country). All control variables, WTC, and WTV were dummy coded using the first category as reference. The variables were included as fixed factors. All dependent measures were scaled from 0 to 100 per cent (0 = low health status/no complaints/no sleep problems/low well-being; 100 = good health status/all psychosomatic complaints/all sleep problems/high well-being). For linear mixed-effects models, unweighted unstandardized regression coefficients (*β*) are reported with standard errors and significance levels based on Satterthwaite approximations for degrees of freedom [74].

In order to gain an impression of whether differences exist between different occupational sectors, the analyses were carried out separately for selected sectors, i.e., 1. hospitality, 2. retail, and 3. health and social care. NACE (See https://nacev2.com/en (last accessed on 14 October 2022)) codes (see Section 2.3) were used to classify the respondents’ occupational sectors as follows: 1. hospitality: NACE section I, divisions 55–56 (accommodation and food service activities), 2. retail: NACE section G, division 47 (retail trade, except of motor vehicles and motorcycles), 3. health and social care: NACE section Q, divisions 86–88 (human health and social work activities).

The models were fitted in two stages. Firstly, only control variables, including countries as random intercepts, were added. Secondly, WTC, WTV, and the interaction of WTC and WTV completed the model. The models were compared by means of analysis of variance (ANOVA) and goodness-of-fit indices (χ²). Random intercept factors (European countries) were compared using intra-class correlations (ICC). It should be noted that both WTC and WTV are correlated (high WTC is associated with high WTV, Fisher’s Exact Test: *p* < 0.001, OR = 3.84, 99.9%-CI) [3.46, 4.26]. In light of the cross-sectional study design, it should be emphasized that the results exclusively describe relationships between the constructs without it being possible to put forward any causal interpretations.

## 3. Results

### 3.1. Descriptive Results

A large number of European employees report low WTC (75%). Only one out of four reports the ability to set their working time arrangements by themselves (25%). WTV is more evenly distributed: nearly half of employees report fixed schedules (low WTV: 48%), while the other half have variable schedules (high WTV: 52%). In combination, working time set by the employer is often accompanied by fixed schedules (low WTC, high WTV: 41%). In contrast, employees who determine their working time themselves report variable schedules more often (high WTC, high WTV: 18%) than fixed schedules (high WTC, low WTV: 7%).

European Countries. WTC and WTV are unequally distributed across Europe (see Appendix A, Table A1 for more details). While WTC is most common in the northern countries, it is rather the exception than the rule in the southern, central-eastern European, and Baltic countries (see Figure 2). Anglo-Saxon and central European countries have slightly higher shares of high WTC. At the country level, Denmark (45%), Sweden (42%), Finland (35%), and the Netherlands (34%) have the highest shares of high WTC/high WTV. By contrast, the lowest rates are found in Lithuania (6%), Romania (6%), Cyprus (4%), and Bulgaria (3%). As the degrees of WTC and WTV differ between occupations, some of the differences between countries might be attributable to differences in their occupational structures.

Occupational sectors. WTV and WTC slightly differ between occupational sectors (see Figure 3). Compared to the overall sample (25%), workers in all three sectors have high WTC marginally less often (hospitality: 19%, retail: 18%, health and social care: 22%). Workers in hospitality (63%) and health and social care (62%) also have high WTV more frequently than is found for the overall sample (52%). The figure for employees in retail (51%) does not differ from that for the overall sample. Low WTC/high WTV is the most common combination for employees in hospitality and health and social care. The figures for the overall sample show that employees are most likely to have low WTC and low WTV, just like employees in the retail sector.

### 3.2. Multi-Level Results

The next step was to analyze the relationship between WTC, WTV, and employee health in several linear multi-level regression models (see Table 2). The low ICC scores indicate that the variation of the outcome variables (health and well-being) across the countries is relatively small (0.02 ≤ ICC ≤ 0.06).

Health. No significant correlation can be found between WTC and self-rated health (see Table 2, *β* = 0.32, SE = 0.465). High WTV seems to be negatively correlated with self-rated health (*β* = −1.92, SE = 0.258). Furthermore, the results indicate no significant interaction between WTC and WTV for self-rated health (*β* = 1.03, SE = 0.549). No significant correlation between psychosomatic health complaints and WTC can be found (*β* = 0.14, SE = 0.526). Employees report significantly more psychosomatic health complaints with high WTV (*β* = 4.53, SE = 0.292). The significant negative interaction of WTC and WTV (*β* = −1.29, SE = 0.621) suggests employees with high WTV report fewer psychosomatic health complaints if they have high WTC than those with high WTV and low WTC.

Sleep problems. Sleep problems are, again, not significantly correlated to WTC (*β* = −0.14, SE = 0.647). However, noticeable sleep problems are significantly related to high WTV (*β* = 4.30, SE = 0.359). The interaction between WTC and WTC is not significant for sleep problems (*β* = −0.80, SE = 0.765).

Well-being. WTC is not significantly correlated with well-being (*β* = 0.15, SE = 0.519). However, WTV is significantly negatively related to well-being (*β* = −3.85, SE = 0.288). The interaction effect of WTC and WTV on well-being is not significant either (*β* = 0.52, SE = 0.613).

### 3.3. Robustness Checks

In order to check the sensitivity of the results, a series of robustness checks were conducted using stratified samples and subgroups (see Appendix B, Table A2 and Table A3). One might argue that employees who use information and communication technology (ICT) have more autonomy and WTC. Therefore, first the analysis was restricted to employees who often worked with ICT (Employees who use ICT at least half of their working time, including “computers, laptops, smartphones etc.” (Q30i)). More than one in two employees (54%) work at least half of their working time with ICT. For similar reasons, general autonomy at work (Employees with the “ability to choose the order or tasks and methods of work” (Q54a and Q54b)) was included in a second analysis. 

The next step was to control for organizational factors relevant to employee representation, as well as safety and health representatives. Three out of four employees (75%) answered at least one of the three questions in the affirmative (This was done using three questions, which asked whether there was (i) employee representation, i.e., a work council, trade union, or similar (Q71a), (ii) a health and safety delegate (Q71b), and (iii) a regular meeting in which employees could express their views on what was happening in the organization (Q71c)). 

WTV is the rule rather than the exception in shift work. Consequently, shift workers were analyzed separately (24 per cent of the sample). All the robustness checks suggested the estimates were quantitatively and qualitatively consistent with the main model, pointing in the same direction and supporting the same conclusion.

### 3.4. Occupational Sectors

Overall, stratified multi-level results checking for differences between occupational groups showed no systematic differences or patterns that diverged from the total sample (see Appendix C). The low ICC scores, comparable to those in the total sample, indicate that the variation of the outcome variables (health, sleep, and well-being) across the countries is relatively small, even between specific sectors (0.02 ≤ ICC ≤ 0.08). Estimate sign changes for WTC or interaction terms could be identified for some models. However, the estimators of both the total sample and the occupational sectors were not statistically significant. Due to the smaller sample sizes in the occupational sectors, the estimates were found to have larger standard errors. For WTV, all estimates show the same sign (same positive or negative association) for the four outcome variables as in the total sample. Although the sizes of the estimators vary slightly across occupational sectors, they are not significantly different from what is found for the overall sample. Hospitality represents an exception with regard to well-being (see Appendix C, Table A4 and Table A5), since a positive association with WTC can be identified in this sector (*β* = 6.74, SE = 2.955). WTV correlates negatively with well-being (*β* = −3.51, SE = 1.171), which is consistent with the total sample. Additionally, a significant interaction is found (*β* = −6.94, SE = 3.385).

## 4. Discussion

This article explores irregular working time arrangements in terms of WTC and WTV based on the EWCS 2015. The descriptive results suggest both aspects are unevenly distributed across the European countries. In the northern countries, employee-oriented flexibility (high WTC) is more common than in the southern, central-eastern, or Baltic countries. As expected, the different distributions of WTC and WTV across the European countries match working time regimes and country clusters [4,20,46], as well as national social welfare regimes [69,75]. Employees benefit more from working time flexibility in the Nordic countries and the Netherlands especially. Furthermore, flexible working hours are deemed more acceptable there for both men and women than in other countries [20,69,75]. In the Nordic countries, statutory legislation sets a general framework for working time standards to be regulated through agreements at a sectoral or company level [76]. In these countries, legislative procedures also involve direct or indirect negotiation with tripartite institutions and social dialogue. This leads to stronger levels of compliance with working time regulations (e.g., the EU Working Time Directive, 2003/88/EC) and smaller deviations from collectively agreed provisions [76].

For European employees, high WTC is less common in hospitality, retail, or health and social care. Hospitality and health and social care workers also have high WTV more often than the overall sample. In contrast to the overall sample, low WTC/high WTV is the most common combination in both hospitality and social and health care. The distribution confirms the importance of addressing irregular working hours in these occupational sectors. Irregular working hours should consequently remain an important aspect of the European agenda for decent working conditions and the efforts to reduce inequality between occupational sectors.

The results of the multi-level model indicate that high WTV is associated with worse health (lower self-rated health status, more psychosomatic complaints), more sleep problems, and poorer well-being compared to low WTV. However, only one significant interaction indicates that WTC and WTV slightly interact: if variability is driven by employees (high WTC), the negative relationships between psychosomatic health complaints and WTV are attenuated. However, WTC does not fully compensate for the negative effect of WTV on health complaints—the positive effect size of the interaction is smaller than the negative effect size of WTV. WTV seems to be a key characteristic of irregular working hours. It reflects both atypical working hours and the disruption of social and biological rhythms, not only for shift work but also for a great many flexible working time patterns. Irregularity often goes hand in hand with unpredictability and precarious working conditions.

Large differences are also evident between occupational sectors. WTV occurs significantly more often among employees in hospitality and health and social care than in the total sample. High WTC is also slightly less common in hospitality, retail, or health and social care. These differences confirm findings about occupational working time patterns in these sectors (see Section 1). The patterns uncovered reflect the great irregularity of working hours due to shift work and externally determined working hours, particularly in hospitality and health and social care [22,63]. High numbers of lower-skilled employees work in hospitality and retail especially. These employees have faced a steeper increase in job strain than higher-skilled workers in recent years [77]. However, the correlations between WTC and WTV and health or well-being remain largely the same for different occupational sectors. A negative correlation to health, sleep, and well-being could be identified in all three occupational sectors. No significant association with WTC or interaction of WTC and WTV could be found, implying that WTV is not only a burden for specific groups of employees, but a general challenge for health and well-being at work. One exception was found among employees in hospitality, where there was a significant positive correlation between WTC and well-being and a negative interaction between WTC and WTV.

The results add to the ambiguous findings regarding WTV. Variable, unstable, and irregular working hours are associated with poor health, sleep problems, and lower well-being. Variable working hours conflict more often with biological and social rhythms [53,78,79] and thereby entail a higher risk of disturbed sleep, poor work–life balance, and low well-being [13,31,54].

Some analyses suggest WTC can attenuate the negative association between WTV, health, and well-being. These findings support the role of autonomy in working time arrangements. If variable schedules are unavoidable, e.g., in shift work, WTC can reduce the negative effects of highly variable working schedules. For example, studies show that high WTC in shift work can reduce stress [80], improve sleep and recovery [34,81], and is positively associated with health outcomes [34,82]. These interaction effects were not confirmed in stratified analyses of different occupational sectors, maybe due to the smaller sample sizes.

At first sight, high WTC seems to be an important resource for shift workers who have highly variable schedules. However, Karhula et al. [11] report that shift workers with high control over scheduling do not necessarily choose the most ergonomic schedules. With high WTC, employees are responsible for their working time arrangements and—paradoxically—often make rather unfavorable and unhealthy choices. In line with these results, high WTC can also lead to high levels of distress and burnout in shift workers [83]. Consideration should therefore be given to the interaction of WTV and WTC. WTC (flexibility availability, see [67]) should not be unlimited, but rather unfold within the boundaries of healthy working time arrangements. Standards for safe and healthy working time arrangements, such as minimum rest periods and maximum weekly working hours, are established in the European Working Time Directive (2003/88/EC). These standards limit WTV, but leave employers and employees room for flexibility [84]. Otherwise, WTC turns from a resource into a liability [40]. Future analyses should focus on exploring such turning points for WTC in combination with WTV.

As sleep, health, and well-being are closely related, longitudinal studies should examine the causal relationships between outcomes. For example, sleep may function as a mediator for health and well-being and so explain the mechanism underlying the chronobiologic desynchronization caused by WTV.

In addition to temporal autonomy, spatial autonomy, or “workplace control” (remote working, working from home, mobile working) should also be considered. Especially during the COVID-19 pandemic, WTC and WTV increased significantly with working from home. Both temporal and spatial flexibility dominate the daily work of more and more employees in Europe [85,86]. By contrast, many workers will still have low WTC or are not employed in teleworkable jobs, which can lead to increasing inequality among employees [87]. This is particularly true of the occupational sectors studied here. Many of the jobs and tasks in hospitality, retail, and social and health care are characterized as interactive work, i.e., involve social interactions with customers, clients, patients, or similar third parties [88].

### 4.1. Theoretical Implications

The results also have theoretical implications for the terminology of working time flexibility, WTC, and WTV. Employee-oriented flexibility or WTC per se may have no consequences for working time arrangements if it is only available to the employees, but not used by them [67]. The use of employee-oriented flexibility is effective when it comes to WTV. Working time flexibility should therefore always be linked to its effects on WTV in order to evaluate the ergonomic aspects of flexible and irregular working time arrangements.

### 4.2. Limitations

Firstly, the dataset does not allow longitudinal analyses, and the analyses are only cross-sectional. In consequence, only correlations can be examined, and it is not possible to make causal claims.

Secondly, the data were collected back in 2015. Due to the massive shift to teleworking during the COVID-19 pandemic, European employees’ WTC and WTV have increased significantly [85]. Future research should also consider the implications of the COVID-19 pandemic for irregular and variable work schedules and the massive shift to flexible working hours for large numbers of employees. Nevertheless, the EWCS 2015 is the most recent dataset that can be drawn on, as the 2020 data collection was postponed to 2021 and the data are not available yet. Furthermore, the data from the EWCS about working time characteristics at an international (European) level are quite unique, so no other datasets can be used that allow these specific research questions to be addressed.

Thirdly, all the variables studied were operationalized via self-report. The exclusive use of a single data collection method poses a risk of common-method bias because respondents’ self-reporting is the source of both the independent and dependent variables in the statistical models [89].

Fourthly, the assessment of working time arrangements is based on subjectively reported data only. In contrast to “objective” data derived from electronic working time records, estimates of working hours are subject to certain biases and lead to inconsistent results [52]. Future studies should address methodological biases in relation to WTC and WTV based on register data, working time recording, or time use surveys to explore in depth how employees’ actual working time patterns vary over days and weeks. Furthermore, more detailed scales may be used to assess different aspects of WTC, e.g., [37], not only with regard to working time, but also spatial flexibility and variability of work (i.e., working from home, mobile working, virtual teamwork).

## 5. Conclusions

To summarize, the descriptive results show that European countries differ in terms of WTC in ways that match their existing working time regimes. The multivariate analyses show negative relationships between WTV and health, sleep, and well-being. To some extent, WTC attenuates the negative relationship between WTV and health. Further studies should analyze WTC in greater detail, looking at several dimensions of WTC. The results indicate that it is important to consider both the availability and usage of flexible work arrangements when studying irregular working times. WTC and WTV are distinct but interrelated constructs. Understanding the associations and interactions between both dimensions is important in order to build up a holistic picture of working time arrangements.

## Figures and Tables

**Figure 1 ijerph-19-14778-f001:**
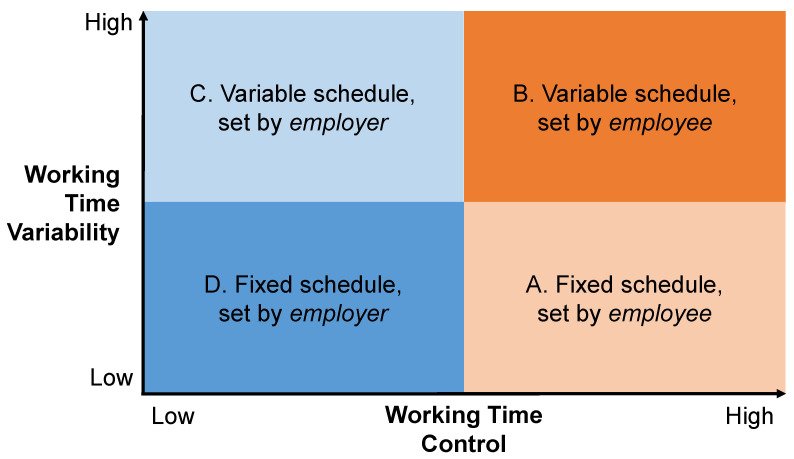
Four types (A.–D.) of working time control and working time variability adapted from [51].

**Figure 2 ijerph-19-14778-f002:**
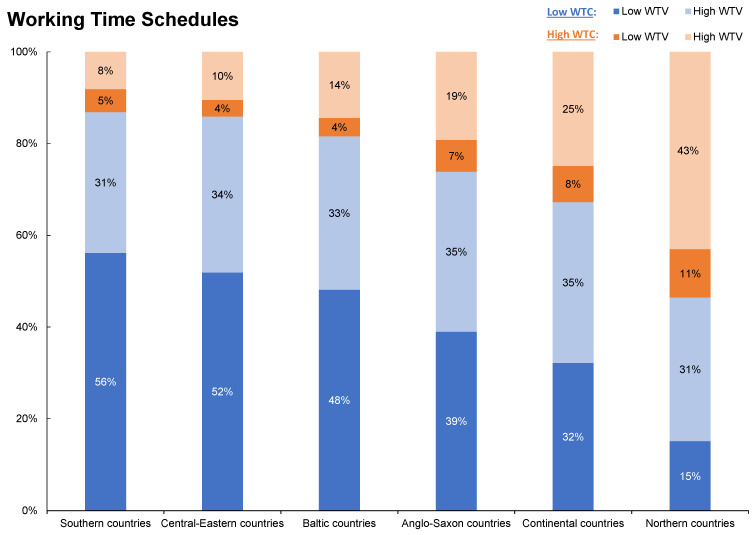
Distribution of working time schedules (WTC and WTV) across European country clusters, weighted by country weights, Anglo-Saxon countries: Ireland, UK; Baltic countries: Estonia, Latvia, Lithuania; central-eastern countries: Bulgaria, Croatia, Czech Republic, Hungary, Poland, Romania, Slovenia, Slovakia; continental countries: Austria, Belgium, France, Germany, Luxembourg, Netherlands; northern countries: Denmark, Finland, Sweden; southern countries: Cyprus, Greece, Italy, Malta, Portugal, Spain; WTC: working time control; WTV: working time variability.

**Figure 3 ijerph-19-14778-f003:**
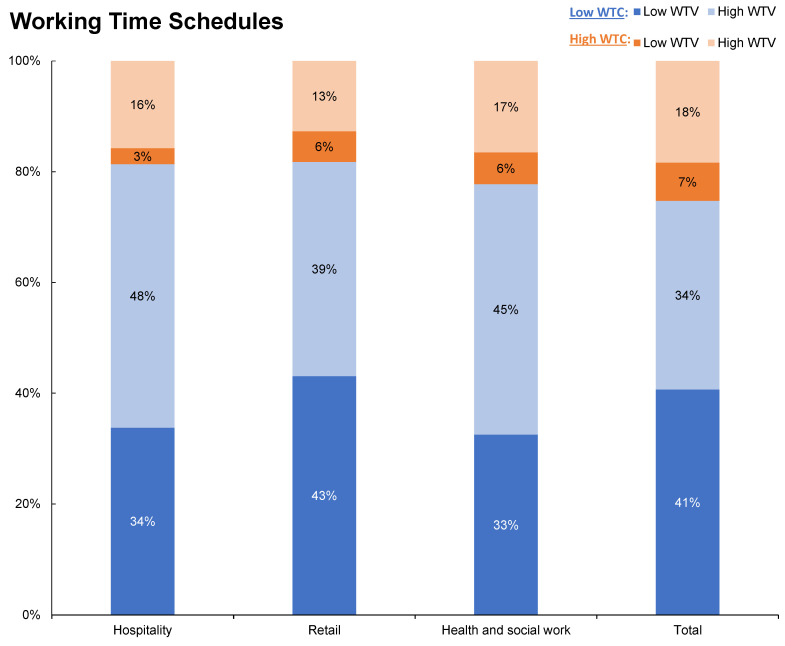
Distribution of working time schedules (WTC and WTV) across selected occupational sectors, weighted by cross-national weights for the EU28, WTC: working time control, WTV: working time variability.

**Table 1 ijerph-19-14778-t001:** Descriptive sample statistics (weighted by cross-national weights for the EU28, unless noted).

		Sample		WTC			
				WTC Low		WTC High	
		% (Column)	N (Unweighted)	WTV Low	WTV High	WTV Low	WTV High
				% (Row)	% (Row)	% (Row)	% (Row)
(1) Individual characteristics							
Sex	Male	50	14,005	39	35	6	20
	Female	50	14,051	42	33	7	17
Age (in years)	15–29	19	5438	38	38	6	18
	30–44	38	10,587	42	34	7	17
	45–54	27	7631	41	34	7	19
	55–65	16	4400	41	30	9	20
Education (ISCED)	Low	16	4387	51	33	5	11
	Medium	52	14,634	44	35	6	14
	High	32	9036	30	33	9	28
Spouse/partner in household	Yes	32	8941	41	36	5	18
	No	68	19,115	40	33	8	19
Age youngest child (<18) in household (in years)	No child	64	18,055	41	35	7	17
	<7	16	4490	37	35	8	19
	7–12	11	3148	42	31	7	21
	13–17	8	2364	39	34	7	21
Household size (no. of individuals)	1	13	3754	39	35	6	20
	2	30	8551	40	34	8	18
	3	24	6829	43	34	6	17
	4	23	6408	40	34	7	19
	5	7	1843	40	32	8	20
	6 or more	2	672	47	30	7	16
(2) Working time characteristics							
Shift work	No	76	21,303	42	28	8	22
	Yes	24	6754	36	53	3	8
Actual working time (hours/week)	1–19	9	2483	31	36	7	26
	20–34	18	5037	37	38	8	16
	35–39	22	6300	40	34	9	17
	40–47	40	11,200	50	29	6	15
	48–59	9	2395	24	43	4	29
	60+	2	642	16	46	3	35
(3) Working time control (WTC) and working time variability (WTV)							
WTC	Low	75	20,976	54	46	–	–
	High	25	7081	–	–	27	73
WTV	Low	48	13,333	86	–	14	–
	High	52	14,723	–	65	–	35
N		100	28,056	41	34	7	18
N (unweighted)		100	27,706	43	33	6	18

**Table 2 ijerph-19-14778-t002:** Multi-level results.

Independent Variables	Dependent Variables (All Scaled 0–100 per Cent)			
	Self-Rated Health (0 Very Bad, 100 Very Good)	Psychosomatic Health Complaints (0 No Complaints, 100 All Complaints)	Sleep Problems (0 None, 100 All Problems)	Well-Being (WHO-5-Scale) (0 Very Low, 100 Very High)
WTC: high WTC ^1^	0.32 (0.465)	0.14 (0.526)	−0.14 (0.647)	0.15 (0.519)
WTV: high WTV ^2^	−1.92 *** (0.258)	4.53 *** (0.292)	4.30 *** (0.359)	−3.85 *** (0.288)
Interaction: high WTC × high WTV	1.03 (0.549)	−1.29 * (0.621)	−0.80 (0.765)	0.52 (0.613)
Control variables for all dependent variables	EU28 countries (random intercept), individual characteristics (gender, age, education level, marital status, age of youngest child ≤18 years in household, household size), occupation, economic sector, working time characteristics (actual working time duration and shift work), see Table 1			
ICC (EU 28 country level)	0.06	0.06	0.05	0.02
χ^2^ (df = 3)	62.0 ***	268.7 ***	165.76 ***	210.8 ***
N	27,706	27,706	27,706	27,706

Note: Unstandardized *β*s, standard errors (SE) in parentheses, WTC: working time control, WTV: working time variability, ^1^ reference: low WTC, ^2^ reference: low WTV, *** *p* < 0.001, * *p* < 0.05, χ^2^ comparison of full model WTC, WTV, and interaction. EWCS 2015, unweighted results.

## Data Availability

The scientific use file of the sixth European Working Conditions Survey 2015 is available at https://beta.ukdataservice.ac.uk/datacatalogue/doi/?id=7363#7 (accessed on 31 August 2022).

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
