# Peer review of "Working Time Control and Variability in Europe Revisited: Correlations with Health, Sleep, and Well-Being"

_ijerph, 2022, doi:10.3390/ijerph192214778_

Round 1
Reviewer 1 Report
Dear Authors,
This study aimed to examine how worktime control and variability of working times are associated with health-related outcomes. The primary findings suggested that WTV is more negatively related to health outcomes compared with WTC. In terms of occupational health, the topic is very important, and the study design sounds good. The sample size is large. However, there are some concerns which listed below. Hopefully, my comments can contribute to sophisticate your manuscript.
Major comments
1. Effect of COVID-19; Given that the data were collected in 2015, I am concerned about the generalizability of this findings. I guess that worktime control would be changed by the spread of new working style, such as remote working. The authors should discuss about the point.
2. Practical implications: It seems to me that the discussion was just academically drawn based on the results. This is not bad, however, practical implications are important for readers because of occupational health research.
3. Shift work leads to worktime variability: I am concerned about whether the negative effects of WTV were derived from shift workers, though the authors controlled for shift work. I understand that the confounding effects statistically adjusted in this model, but shift workers were more popular in WTV high group in Table.1. If possible, more analyses would be beneficial to deny the possibility, like robustness check in Appendix B.
4. Sleep as mediator: Some previous studies focused on sleep. I think that sleep can play a mediator role in determining the combined effects of WTC and WTV on health outcomes. It would be better to add the data of sleeping hours or sleep problems in this analysis (e.g., demographic data or control variables).
Minor comments
1. L297; Is it typo? “Low” WTC?
2. Table.2; Marginally significant effect (p < .10): In general, this practice distorts the results and implies some form of practical significance for what are otherwise non-significant findings. See https://journals.sagepub.com/doi/10.1177/0956797616645672
3. L101; The authors addressed that the combined analysis of WTC and WTV has been rarely conducted, however, the following papers have been conducted. Especially, the finding of Costa et al is line with your finding. The finding would be beneficial to deepen your discussion.
Costa G, Akerstedt T, Nachreiner F, Baltieri F, Carvalhais J, Folkard S, Dresen MF, Gadbois C, Gartner J, Sukalo HG, Härmä M, Kandolin I, Sartori S, Silvério J. Flexible working hours, health, and well-being in Europe: some considerations from a SALTSA project. Chronobiol Int. 2004;21(6):831-44.
Kubo T, Takahashi M, Togo F, Liu X, Shimazu A, Tanaka K, Takaya M. Effects on employees of controlling working hours and working schedules. Occup Med (Lond). 2013 Mar;63(2):148-51.
Thank you for your exciting work.
Author Response
Thank you very much for the opportunity to revise and resubmit our paper. We highly appreciate the time and effort the Editor and the Reviewers have put into reading and commenting our manuscript.
|
Major comments |
Author’s response |
|
1. Effect of COVID-19; Given that the data were collected in 2015, I am concerned about the generalizability of this findings. I guess that worktime control would be changed by the spread of new working style, such as remote working. The authors should discuss about the point. |
Thank you for this comment. We have removed the topic COVID-19 from the introduction to avoid creating false expectations about the article or the data. We discuss the effect of the pandemic in the discussion section (limitation and conclusion). |
|
2. Practical implications: It seems to me that the discussion was just academically drawn based on the results. This is not bad, however, practical implications are important for readers because of occupational health research. |
Thank you for this comment. We added some practical implications in the discussion section and hope that you like them. |
|
3. Shift work leads to worktime variability: I am concerned about whether the negative effects of WTV were derived from shift workers, though the authors controlled for shift work. I understand that the confounding effects statistically adjusted in this model, but shift workers were more popular in WTV high group in Table.1. If possible, more analyses would be beneficial to deny the possibility, like robustness check in Appendix B. |
Thank you for this valuable advice. We ran an analysis regarding in the robustness check systems separately for shift workers and non-shift workers. The analysis is included in appendix B. Basically; the results do not differ for shift workers. |
|
4. Sleep as mediator: Some previous studies focused on sleep. I think that sleep can play a mediator role in determining the combined effects of WTC and WTV on health outcomes. It would be better to add the data of sleeping hours or sleep problems in this analysis (e.g., demographic data or control variables). |
Thank you for your comments on sleep as a mediator. Regrettably, the EWCS does not include sleep hours in the dataset, however, sleep problems are included (3-item scale) and we added sleep as an outcome variable. However, we only discuss the potential of sleep as a mediator, since mediation analyses in cross-sectional data is widely criticized (e.g. Maxwell, S. E., Cole, D. A., & Mitchel, M. A. (2011). Bias in cross-sectional analyses of longitudinal mediation. Partial and complete mediation under autoregressive model. Multivariate Behavioral Research, 46, 816–841. https://doi.org/10.1080/00273171.2011.606716; Maxwell, S. E., & Cole, D. A. (2007). Bias in cross-sectional analyses of longitudinal mediation. Psychological Methods, 12(1), 23–44. https://doi.org/10.1037/1082-989X.12.1.23)
|
|
Minor comments |
|
|
1. L297; Is it typo? “Low” WTC? |
The typo has been corrected (“High” WTC). Thank you for your advice. |
|
2. Table.2; Marginally significant effect (p < .10): In general, this practice distorts the results and implies some form of practical significance for what are otherwise non-significant findings. See https://journals.sagepub.com/doi/10.1177/0956797616645672
|
Thank you for your comments regarding this methodological issue and the interesting article you recommended to read. In order to avoid the increased prevalence of findings that provide weak evidence, I removed the term “marginally significant” from the results section and removed also the threshold of .10 in table. |
|
3. L101; The authors addressed that the combined analysis of WTC and WTV has been rarely conducted, however, the following papers have been conducted. Especially, the finding of Costa et al is line with your finding. The finding would be beneficial to deepen your discussion. |
Thank you for the two references – they were already included in the manuscript at another paragraph in the text. However, Costa et al. analyze WTC and WTV in a different way with separate estimators without interaction. So the analysis with an interaction is quite new / unusual. The analysis of Kubo is quite similar but we enlarged the analysis of Kubo to another dataset and used another methodology (multi-level analyses). In general, our analysis goes in line with Kubo’s idea and results. |
|
Thank you for your exciting work. |
Thank you, again, for your nice words and all your highly constructive and helpful comments. |
Reviewer 2 Report
Thank you for providing an opportunity to review the manuscript 'Working time control and variability in Europe revisited: Correlations regarding health and well-being”. I liked to read the manuscript and found it promising. However, I am concerned about the contribution this manuscript can make. Consequently, I have made the following suggestions to improve the quality of current work:
First, the author should explain why the 2015 data are relevant and the analysis should be interesting and published in 2022. As we know from other studies, the pandemic has had a significant impact on working time variability, so it is quite unclear why prepandemic data are used, despite its completeness and its international comparability.
Second, the data analysis should be extended. The paper provides information that there are differences between the country groups (in Figure 2 and the text), but I miss the analysis of how the country/country group factor works when analyzing the impact of WTC and WTV impact on employee health.
I hope the feedback is useful. Thanks.
Author Response
Thank you very much for the opportunity to revise and resubmit our paper. We highly appreciate the time and effort the Editor and the Reviewers have put into reading and commenting our manuscript.
|
Major comments |
Author’s response |
|
Thank you for providing an opportunity to review the manuscript 'Working time control and variability in Europe revisited: Correlations regarding health and well-being”. I liked to read the manuscript and found it promising. |
Thank you for your review and your helpful comments. It is very nice to hear that you generally approve of the manuscript. |
|
However, I am concerned about the contribution this manuscript can make. Consequently, I have made the following suggestions to improve the quality of current work: First, the author should explain why the 2015 data are relevant and the analysis should be interesting and published in 2022. As we know from other studies, the pandemic has had a significant impact on working time variability, so it is quite unclear why prepandemic data are used, despite its completeness and its international comparability. |
I discuss why we used this dataset in the discussion section. Even though the data is from 2015 it should be taken into account that this is still the most recent EWCS data, as the 2020 data collection has been postponed and the data is not yet published. We have removed the topic “COVID-19” from the introduction to avoid creating false expectations about the article or the data. We discuss the effect of the pandemic in the discussion section (limitation and conclusion).
|
|
Second, the data analysis should be extended. The paper provides information that there are differences between the country groups (in Figure 2 and the text), but I miss the analysis of how the country/country group factor works when analyzing the impact of WTC and WTV impact on employee health. I hope the feedback is useful. Thanks. |
Since the country (groups) were different as to the extent of WTC and WTV and its combination, this has little impact on the results regarding the correlation of WTC and WTV on health and well-being (see small ICCs).
Thank you, again, for your nice words and all your highly constructive and helpful comments. Your feedback was indeed very helpful. |
Round 2
Reviewer 1 Report
Dear Authors,
Thank you for revising your manuscript. I confirmed that your manuscript has been revised according to my previous comments.
I hope that your findings will contribute to improve health at work. I look forward to your next work.
Author Response
Thanks again for the helpful hints and comments. I am glad that all changes have been implemented to your satisfaction.
Reviewer 2 Report
The author has improved the paper and responded to all the recommendations made.
Author Response

(The authors gave the same response as above.)
